# When Do Decompositions Help for Machine Reading?

**Kangda Wei[1], Dawn Lawrie[2], Benjamin Van Durme[2], Yunmo Chen[2*], Orion Weller[2*]**
[1]Department of Computer Science and Engineering, Texas A&M University
[2]Department of Computer Science, Johns Hopkins University
kangda@tamu.edu, {lawrie, vandurme, yunmo, oweller2}@jhu.edu

## Abstract

Answering complex questions often requires multi-step reasoning in order to obtain the final answer. Most research into decompositions of complex questions involves open-domain systems, which have shown success in using these decompositions for improved retrieval. In the machine reading setting, however, work to understand when decompositions are helpful is understudied. We conduct experiments on decompositions in machine reading to unify recent work in this space, using a range of models and datasets. We find that decompositions can be helpful in zero or limited-data settings, giving several points of improvement in exact match. However, we also show that when models are given access to around a few hundred or more examples, decompositions are not helpful (and can actually be detrimental). Thus, our analysis implies that models can learn decompositions implicitly even with limited data. [1]

## 1 Introduction

Much past work has examined and improved models' ability to answer complex questions that require multiple steps of reasoning (Welbl et al., 2018; Talmor and Berant, 2018a; Dua et al., 2019a; Wolfson et al., 2020; Weller et al., 2020; Weir and Van Durme, 2022). A consistent theme in these works is to break the main complex question down into a series of sub-questions to be solved, which is referred to as question decomposition. These methods generally represent decompositions as a human would, with explicit natural language sub-questions that build together to the final answer.

Despite the large amount of research in decompositions for multi-step question answering, the majority of it has focused on using question decomposition for both information retrieval and question

---

[1]We publicly release all code at https://github.com/WeiKangda/Question-Decomposition

  \* Joint advising

**No Decomposition**
Who co-founded View Askew Productions and produced numerous movies starring Jason Lee?

**Explicit Decomposition**
1. Who co-founded View Askew Productions?
2. Who produced numerous movies with Jason Lee?
3. Who is in both #1 and #2
↳ Final Answer

**Implicit Decomposition**
`SELECT` `INTERSECTION` Who co-founded View Askew Productions and produced numerous movies starring Jason Lee?

Figure 1: An instance of HotpotQA in BREAK (Wolfson et al., 2020), showing three different decomposition settings: (1) No Decomposition, i.e. regular question answering, (2) Explicit Decompositions that use iterative sub-questions, and (3) Implicit Decompositions that prepend the reasoning steps as special tokens.

answering (e.g. in the open domain). In that setting, results have consistently shown that question answering (QA) models perform better on multi-step questions when they use decomposed questions to help with retrieval (Wolfson et al., 2020; Perez et al., 2020; Geva et al., 2021a). The few works that have used decompositions for machine reading do so in limited settings, leaving its effectiveness unclear (Guo et al., 2022; Patel et al., 2022).

Therefore, we seek to shed light on **if and when** decompositions are helpful for machine reading. To do so, we analyze decomposition methods for several QA models across two multi-step QA datasets. Our results show that decompositions are only helpful in the low data setting, where there are less than a few hundred examples. Using decompositions in anything other than that setting performs the same (or much worse, depending on the strategy) as simply training the model end-to-end.

Thus, overall, decompositions are helpful for question answering when they are used in two main

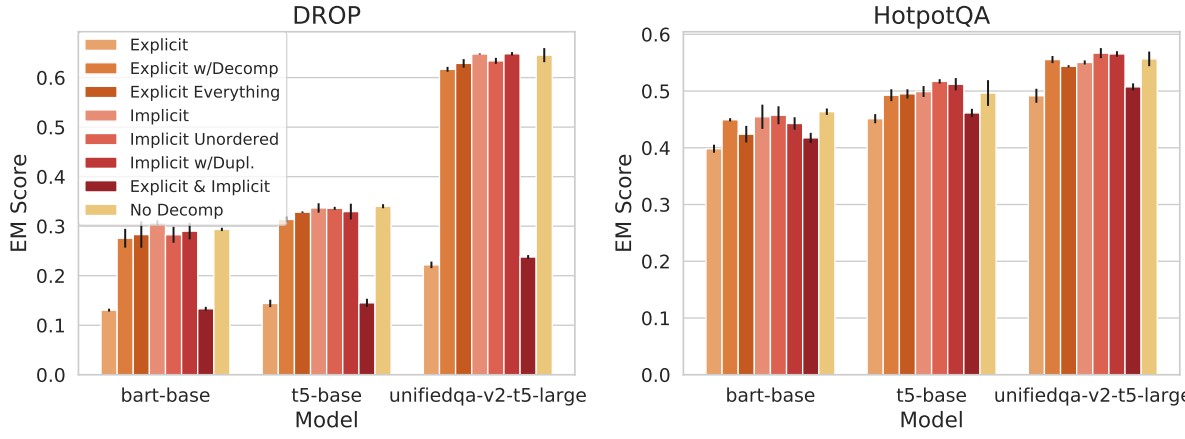

Figure 2: Main results showing the effect of various decomposition strategies. Left shows results on DROP, right shows HotpotQA. Runs were done over three random seeds and report the mean; error bars indicate 1 SD.

settings: (1) information retrieval, where the decompositions help isolate distinct aspects of the question, or (2) in zero to low resource settings, where there isn't enough data to implicitly learn the multi-step process through end-to-end training.

## 2 Experiment Setup

### 2.1 Data

We use the BREAK (Wolfson et al., 2020) resource which contains annotated decompositions for 10 different benchmarks, including three reading comprehension benchmarks. BREAK provides human-written decompositions for all dataset splits (e.g. train, test) which we use, thus assuming the oracle decomposition case. We use two of these three datasets (HotpotQA and DROP) as the third, ComplexWebQuestions (Talmor and Berant, 2018b), does not currently provide a training set.[2] Following BREAK and other follow-up work (Geva et al., 2021b), we train and test our models using the high-level decompositions (also known as QDMRs).

Thus, we use HotpotQA (Yang et al., 2018) and DROP (Dua et al., 2019b) in our experiments, using the portions annotated from their train and validation sets. HotpotQA was created from workers on Mechanical Turk who annotated compositional questions from Wikipedia by using the page links (an example can be found in Figure 1). DROP was created to test discrete multi-hop reasoning and was also annotated by Mechanical Turk workers who used Wikipedia articles as the context.

---

[2]Note that although the ComplexWebQuestions (CWQ) paper initially released the training set the authors have since removed it from the official Dropbox. Furthermore, even if the dataset was available CWQ does not verify that the questions are answerable from the returned Google search passages.

The BREAK annotations include the list of decomposed questions that eventually yield the same answer as the original question, along with an operator assigned to each decomposition that represents the type of reasoning for that step (e.g. Boolean, Comparison, etc.). Note that there are no gold labels for intermediate decomposition steps; the only ground truth label is for the main question.

### 2.2 Models

To explore the effect of decompositions on various types of common NLP models, we employ five different models: BART (Lewis et al., 2020), vanilla T5 (Raffel et al., 2020), UnifiedQA-v2 (Khashabi et al., 2020, 2022) which uses a T5 backbone and has been fine-tuned on other QA datasets but not on HotpotQA (Raffel et al., 2020), LLaMA-7B (Touvron et al., 2023) and Alpaca-7B (Taori et al., 2023). This allows us to demonstrate the effect of additional fine-tuning (UnifiedQA vs vanilla T5), different architectures (BART vs T5 vs LLaMA), and instruction-tuning (LLaMA vs Alpaca). Note that because UnifiedQA-v2 was multi-task trained on DROP, its scores on it are noticeably higher than the other models (Figure 2). However, our purpose is not to compare scores between models, but rather to compare scores *between different decomposition strategies*. Thus, the inclusion of this model on DROP shows us that our results hold even if the model was pretrained on it. For more hyperparameter and compute details, see Appendix A.

### 2.3 Decomposition Strategies

There are many possible ways to combine decomposition with model fine-tuning. We try a wide variety of techniques (including novel ones) that

we group into three categories: (1) no decomposition e.g. the baseline QA format, (2) explicit decomposition, and (3) implicit decomposition.

**Explicit Decomposition** Explicit decompositions are the most common approach in the decomposition literature, generating the answer iteratively through sub-questions: the model answers the first decomposition step, then replaces placeholders in future decomposition steps with that predicted answer, then predicts the second decomposition step, and so forth. Note that using this method (*Explicit*) naively presents issues with backpropagation, as the model can only backpropagate through the last decomposition step. Variations of strategies in this category include giving the model all previous decomposition steps as context (*Explicit w/Decomp*) or including all decomposition steps and all predicted answers as context (*Explicit Everything*).

**Implicit Decomposition** Another way to use decompositions could be to add them implicitly. To do so, we utilize the *operators* provided in the BREAK annotations which describe the type of reasoning needed, removing duplicate operators and keeping them in their given order. For example, in Figure 1 the model uses *select* twice and then *intersection* in the explicit decomposition reasoning steps (but we remove the duplicate *select*). We implement this in practice by adding a new special token for each operator and prepending them to the original question. Although this approach is novel in the context of decompositions, it bears similarity to work in the prompting literature, such as soft prompts (Qin and Eisner, 2021; Liu et al., 2022). Variations to this approach in the same category include randomizing the order of the special tokens (*Implicit Unordered*), leaving in duplicate special tokens (*Implicit w/Dupl.*), or even prepending these operators to the explicit sub-questions in the Explicit Decomposition approach to combine the two strategies (*Explicit + Implicit*).

## 3 Results

**Full-Data Experiments** We see the main results in Figure 2 with results for DROP on the left and HotpotQA on the right. Bars are colored-coded according to their method. All bars are the mean of three random seeds and error bars indicate the standard deviation. We see that most methods perform nearly the same, except for two that underperform: *Explicit* and *Explicit + Implicit*. Note that both of

| Method | Alpaca | LLaMA |
|---|---|---|
| Explicit | $54.8_{(0.4)}$ | $53.4_{(3.5)}$ |
| Explicit w/Decomp | $53.8_{(2.0)}$ | $57.7_{(0.3)}$ |
| Explicit Everything | $58.7_{(1.3)}$ | $55.9_{(0.5)}$ |
| Implicit | $57.0_{(1.6)}$ | $60.9_{(0.9)}$ |
| Implicit Unordered | $58.3_{(0.7)}$ | $50.9_{(0.6)}$ |
| Implicit w/Dupl. | $58.5_{(2.1)}$ | $53.7_{(0.6)}$ |
| Explicit & Implicit | $55.0_{(0.7)}$ | $59.3_{(0.7)}$ |
| No Decomp | $\mathbf{59.9_{(0.6)}}$ | $\mathbf{59.8_{(1.2)}}$ |

Table 1: Results of using larger LLMs, Alpaca and LLaMA, on HotpotQA. *No Decomp* still performs the same (within two standard deviations) or better when comparing to methods that use decompositions.

these have issues with training end-to-end, as an *Explicit* decomposition is not differentiable through all decomposition steps. Thus, we only end up differentiating through the last step of the explicit decomposition steps, leaving the model unable to learn as effectively. All other approaches to decomposition perform comparably, given random seed variance (e.g. t5-base DROP *Implicit Decomp.* is 33.7% exact match ± 0.9% vs *No Decomp* 34.0% ± 0.4%). In fact, in this full data setting, the *No Decomp* method performs better or statistically similar to every other method according to two-sample t-tests with the Bonferroni correction (Weisstein, 2004), across all datasets and models.

We show the two more recent and larger models (LLaMA and Alpaca) in Table 1 on Hotpot, reporting the mean and standard deviation of the EM score. We find that *No Decomp* method still performs similarly or outperforms all other decomposition methods (e.g. for Alpaca 59.9 EM with *No Decomp* vs 54.8 EM for *Explicit Decomp*). Thus, the conclusion that decompositions are helpful only when there is not enough labeled examples still holds for newer and larger models and even for instruction-tuned models like Alpaca.

**Size Experiment** Is the *No Decomp* method always the same or better than the decomposition methods, or can decompositions help in the low-data regime? We answer this question in Figure 3, by varying the amount of training data, comparing it to the zero-shot *Explicit* method that is typically used (Patel et al., 2022; Dua et al., 2022).[3] As we need the zero-shot model to be fine-tuned on

---

[3]Note that fine-tuned *Explicit* is worse than the zero-shot version in low-data settings (See Figure 3b) We do not evaluate on DROP as UnifiedQA-v2 was already trained on DROP.

QA to handle the decomposed questions,[4] we use a modified version of T5-base (valhalla/t5-base-squad from Huggingface) that was fine-tuned on SQuAD (Rajpurkar et al., 2016). This places the *No Decomp* method at a disadvantage, as it was not trained on SQuAD and started from T5-base. Despite that, however, we see that as the amount of data increases, the fine-tuned models perform better; eventually surpassing the zero-shot method between 100-250 examples for UnifiedQA-v2 and 250-1000 examples for T5.[5]

**Error Analysis**   Why does the zero-shot *Explicit* method perform worse than the fine-tuned *No Decomp*? In Table 2 we show representative errors from the *Explicit* method with how often they occurred. We randomly sample 50 errors and categorize them into three groups: wrong predictions in the last step, error propagation from intermediate steps, and invalid/missing annotations from BREAK (i.e. not the model's fault). We found that the biggest category was predicting an invalid annotation (42%), i.e. an alias that the dataset did not contain, followed by error propagation (40%) and then wrong last predictions (18%). Thus, compared to the non-iterative methods, the iterative process allows error propagation that occurs in roughly 40% of errors, contributing to its lower comparative scores (see Appendices B and C for other error analyses on *No Decomp* and cases where *No Decomp* was better than *Explicit*).

## 4   Related Work

**Decompositions in QA**   Decompositions for QA have a long history in complex question answering (Perez et al., 2020; Min et al., 2019; Geva et al., 2021a) with recent interest in using them for large language models (Wei et al., 2022; Dua et al., 2022; Zhou et al., 2023; Press et al., 2023). Two of the most related works to ours include Patel et al. (2022), who focus on decompositions in the zero-shot setting only and show improvements (which aligns with our results), and other work (Guo et al., 2022) that shows that decompositions help on the full DROP dataset but which doesn't include a comparable non-decomposition baseline on the same data. Our analysis complements and

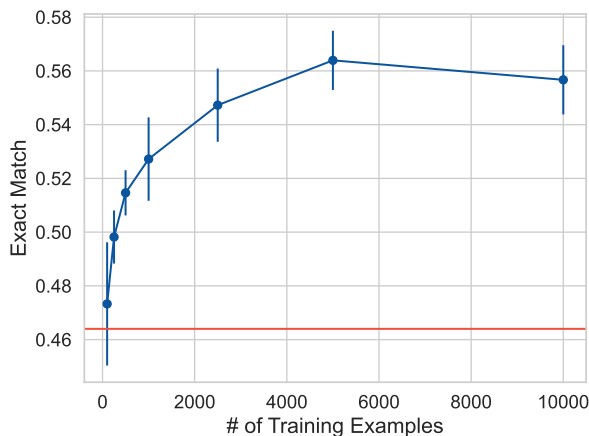

(a) Results from UnifiedQA-v2

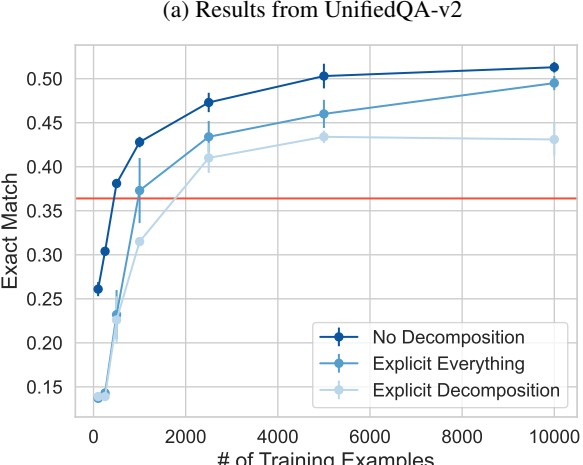

(b) Results from T5-base

Figure 3: Size Experiments w/HotpotQA. For Figure 3a with UnifiedQA-v2, the red line shows the score of zero-shot *Explicit* while the blue line is the fine-tuned *No Decomp* method. For Figure 3b, the red line shows the score of zero-shot *Explicit* from valhalla/t5-base-squad while the blue lines are fine-tuned T5-base with *No Decomp*, *Explicit Everything*, and *Explicit* methods. Results are over 3 random seeds and show the mean and 1 SD. See Appendix D for similar results with Alpaca.

unifies our understanding of decompositions by identifying when decompositions help with respect to dataset size.

**Decompositions in other fields**   Our results for decompositions in textual QA helps to unify results across machine learning, as similar conclusions (e.g. decompositions being less effective then end-to-end methods with large data) can be seen from scores on Computer Vision visual QA leaderboards (Hudson and Manning, 2019; Li et al., 2021).

**Decomposition Strategies and Prompting**   Decompositions methods are also related to prompting, where the explicit decompositions can be seen as a hard prompt (Liu et al., 2021; Su et al., 2022)

---

[4]We use the same UnifiedQA-v2 model for both zero-shot and fine-tuned as it was already trained on QA.

[5]We also show several other decomposition methods over various amounts of fine-tuning in Figure 3b, all starting from T5-base. We see that they perform worse than the *No Decomp* method at every point.

| Question | Decompositions | Content (shortened) | Intermediate Predictions | Answer | Error |
|---|---|---|---|---|---|
| Who was born first, Kwok Kin Pong or Edison Chen? | #1: when was Kwok Kin Pong born? #2: when was Edison Chen born? #3: which is the lowest of #1,#2? | Edison Koon-hei Chen (born 7 October 1980) .. Kwok Kin Pong (born 30 March 1987 in Hong Kong) .. | #1. 30 March 1987 #2. 7 October 1980 #3: Kwok | Edison Chen | Wrong Prediction at Last Step (18%) |
| Are both Deerhunter and Nine Lashes American Christian rock bands? | #1. is Deerhunter a American Christian rock band? #2. is Nine Lashes a American Christian rock band? #3: if both #1 and #2 are true | .. Nine lashes is an American Christian rock band .. Deerhunter is an American rock band from Atlanta .. | #1. Yes #2. Yes #3. Yes | No | Error propagation (40%) |
| What actor from "Willow" also starred in "The Usual Suspects"? | #1: who is the actor that starred in The Usual Suspects? #2: #1 that was a actor from Willow? | .. Kevin Elliot Pollak .. a role in "Willow" .. the Usual Suspects stars Kevin Pollak .. | #1. Kevin Pollak #2. Kevin Pollak | Kevin Elliot Pollak | Invalid or Missing Annotation (42%) |

Table 2: Error analysis for zero-shot decomposition (i.e. *Explicit* with no fine-tuning, a la Patel et al. (2022)) on HotpotQA. Percentages calculated from annotation of 50 instances with representative examples shown. See Appendix B for full data *No Decomp* or Appendix C for when *No Decomp* succeeded and *Explicit* failed.

and the implicit decompositions are similar to soft prompts (Qin and Eisner, 2021; Liu et al., 2022).

Other research has looked at developing new prompting methods which either better handle complex questions or automatically generate decompositions (as opposed to the human-written decompositions in BREAK) (Zhou et al., 2022; Weller et al., 2022, 2023; Press et al., 2022). Our work focuses only on human written decompositions, which were shown to be better than automatically generated decompositions in the BREAK paper.

## 5 Conclusion

Our work explored when decompositions are helpful for machine reading. We showed that decompositions are helpful when there is limited data available, or when parameters cannot be tuned. However, when enough data exists (empirically around a few hundred instances) and parameters can be fine-tuned, it is best to let the model learn the decompositions implicitly through end-to-end training. Furthermore, we show that limitations of not fine-tuned decomposition approaches include error propagation of intermediate steps while also introducing more possibilities for annotator error. We hope that our work will help to inform readers as they create new datasets and select methods to use for complex question answering.

## 6 Limitations

Our work has explored the machine reading setting, using what is to our knowledge the only large com-

plex question answering datasets that have human-annotated decompositions. However, it is possible that in the future someone could create another decomposition-based dataset that could show slightly different results. We believe this to be unlikely, as our empirical study holds across models and datasets. Another limitation is that we do not use alternate tasks, such as summarization, tagging, etc. as our work focuses on how decompositions work in question answering only, given the large interest in using decompositions for QA.

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

# A Hyperparameter and Compute Details

We train our models following the hyperparameters in Wolfson et al. (2020), e.g. a learning rate of 1e-5 for 5 epochs, using early stopping on a holdout of the training set (5%) to determine the best saved model. The best models were typically ones trained for around 2 epochs. We use T5-base, BART-base, UnifiedQA-V2-Large, LLaMA 7B, and Alpaca 7B.

For the zero-shot decomposition UnifiedQA-v2 approach (perhaps due to its multi-task pre-training on other QA datasets), we found that using the original question plus the iterative decompositions

added a solid boost to final performance and hence we use those results as it provides the strongest zero-shot baseline. Each zero-shot decomposition run took approximately 15-30 minutes to evaluate.

Compute time ranged from 1 hr for the shortest jobs with smaller data sizes and non-iterative training to around 18 hours for iterative decompositions methods with UnifiedQA-v2-large on 1 RTX 6000 GPU. For LLaMA and Alpaca, we use one 80GB A100 GPU with runs taking between 12-24 hours.

Data was prepared building off of the original BREAK authors' code. Models were accessed via the Huggingface repository (Wolf et al., 2019).

Note that our reported no-decomposition results on DROP with T5 show comparable or greater performance to Guo et al. (2022) when using only the BREAK dataset.

## B  Error Analysis for Full-Data *No Decomp*

We also do an error analysis for the full-data version of the *No Decomp* method to compare with the *Explicit* error analysis done in the main paper. As error propagation is not a possible category for this model, since it is end-to-end, there are only two error categories we use: (1) Wrong Prediction (54%) and (2) Invalid or Missing Annotation (46%). We see that a missing annotation/alias was the cause of the incorrect answer 46% of the time, which is comparable to the 42% of the time the zero-shot *Explicit* method had an error due to an alias. For the other 54%, the model would output "yes" when it should be "no" or extract the wrong string from the passage leading to an incorrect prediction, etc.

## C  Error Analysis With *No Decomp* succeeded and zero-shot *Explicit Decomp* failed

Based on Table 2, the Invalid or Missing Annotation takes up the largest percentage of error, which could happen to both decomposition and none-decomposition method. In order to further strengthen our point that question decompositions cause error propagation, thus underperforming the *No Decomp* method overall, we also conduct an error analysis where the fine-tuned *No Decomp* succeeded and the zero-shot *Explicit Decomp* failed, with 20 sampled errors using UnifiedQA-v2. The new results are: (1) Error Propagation (45%), (2) Wrong Prediction at Last Step (35%), and Invalid or Missing Annotation (20%). We see similar re-

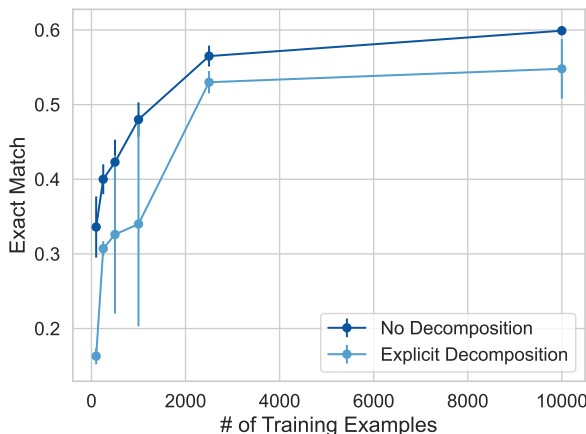

Figure 4: Size Experiments w/HotpotQA using Alpaca. *Explicit* and *No Decomp* results are shown. Results are over 3 random seeds and show mean and 1 SD.

sults to the error analysis in Table 2 where Error Propagation causes the most errors.

## D  Size Experiment for Alpaca

We also performed the size experiment in Section 3 with Alpaca-7B (chavinlo/alpaca-native) and show the results in Figure 4. Note that Alpaca is roughly 10 points better than T5-base (approximately 60 EM vs 51 EM). The *No Decomp* method performs similarly or better than the *Explicit* method in all cases with varying training data size. Our conclusion still remains across models and decomposition strategies: *No Decomp* is better than or similar to the *Decomp* models, when given enough training examples.