# OpenReview forum: "When Do Decompositions Help for Machine Reading?"
_EMNLP/2023/Conference — EMNLP 2023 Main_

### Official Review · Reviewer_dFMJ · 2023-08-02

**Soundness:** 3

**Excitement:**

3: Ambivalent: It has merits (e.g., it reports state-of-the-art results, the idea is nice), but there are key weaknesses (e.g., it describes incremental work), and it can significantly benefit from another round of revision. However, I won't object to accepting it if my co-reviewers champion it.

**Missing References:**

Min, Sewon, et al. "Multi-hop Reading Comprehension through Question Decomposition and Rescoring." Proceedings of the 57th Annual Meeting of the Association for Computational Linguistics. 2019.

- The above paper explains the types of multi-hop questions (especially in HotpotQA), so it may be helpful to understand Implicit decomposition.

**Paper Topic And Main Contributions:**

This paper addresses the question of whether decomposition is helpful in complex question answering (multi-step/hop QA), and if so, under what circumstances. The authors claim that question decomposition is useful when only handful of training samples (multi-hop QA instances) are given, and unhelpful or even hindering when sufficient data are available.

**Questions For The Authors:**

- In lines 165-171, the authors explained why the Explicit and Explicit + Implicit methods didn't work well (problem of Explicit method). Then, couldn't we use variation (Explicit w/Decomp or Explicit Everything) instead of Explicit method in Explicit + Implicit method?

**Reasons To Accept:**

- The writing is generally neat, and the paper is easy to read/understand.

- The conclusions presented in the paper can be used as a guideline for other researchers which can be helpful to the community.

- Experiments are conducted on three models and two datasets to show that their claims are not limited to a specific model or dataset.

**Reasons To Reject:**

- I would like to raise a concern on the size experiment of Figure 3. The authors use QA models fine-tuned on SQuAD (single-hop QA dataset) to compare zero-shot Explicit method and No Decomp method. However, as the question decomposition methods decompose a multi-hop question into single-shop questions, it is rather natural that the zero-shot Explicit method is advantageous when the training dataset is small. In my opinion, this experiment should have been based on the pretrained LMs rather than the SQuAD fine-tuned QA models, comparing the performance of the Explicit method (not zero-shot) and the No Decomp method according to the number of training samples.

- (Suggestion rather than a reason to reject) Since the question decomposition itself provides an additional inductive bias, it can be expected that the training sample efficiency will naturally occur. In order for this paper’s conclusion to be more meaningful, it would be good to add an analysis of the cause. For example, does an ability to decompose a question arise within a QA model given sufficient training data (multi-hop QA instances)? If so, does this ability correspond to the inductive bias provided by the Explicit/Implicit methods?

- (Minor) The presentation of some figures needs to be improved (Refer to Presentation Improvements below).

**Reproducibility:**

5: Could easily reproduce the results.

**Reviewer Confidence:**

5: Positive that my evaluation is correct. I read the paper very carefully and I am very familiar with related work.

**Typos Grammar Style And Presentation Improvements:**

- In Figure 2, the meaning of each bar can be figured out in the legend order, but the colors are too similar. It would be nice to have more color separation for clarity.

- In the case of UnifiedQA-v2 in Figure 3, it is hard to see a better range for the zero-shot Explicit method. Can you provide data or adjust the x-axis of the figure to show it clearly?

- (Line 220) Guo et al. (2022) was published in December 2022, and it may be difficult to call it concurrent due to the time difference.

---

> ### Author Rebuttal · Authors · 2023-08-28
>
> Thank you for your time and for recognizing that our work “addresses an under-explored area in the QA/decomposition field”!
>
> ---
> > I would like to raise a concern on the size experiment of Figure 3 ... In my opinion, this experiment should have been based on the pretrained LMs rather than the SQuAD fine-tuned QA models
>
> **We agree with this sentiment and have already done so!** We used two different models to conduct the size experiment: one is UnifiedQA-V2, which is fine-tuned on SQuAD (and many others), but the other one is base T5, which is a pretrained LM (which is what you’re suggesting). We showed that our conclusion holds in both cases (with the LM fine-tuned on QA datasets and not fine-tuned on QA).
>
> ---
>
> > In order for this paper’s conclusion to be more meaningful, it would be good to add an analysis of the cause ... does an ability to decompose a question arise within a QA model given sufficient training data?
>
> We definitely agree that the decomposition provides an inductive bias and that it is helpful to have an analysis of the cause - in fact **we provide this in the Size Experiments** (Figure 3). These charts show how the scores improve when given N training examples of multi-hop QA instances, just as you suggest (both for UnifiedQA which has seen QA and for T5-base which has not).
>
> If you mean that our result is not surprising, we would agree but mention that the community does not agree, see *“Is a Question Decomposition Unit All We Need?”* by Patel et. al. 2022 and others cited in our paper that suggest question decompositions are better. Hence why we wrote this paper to inform the community since others don’t agree and to provide evidence.
>
> ---
> > couldn't we use variation (Explicit w/Decomp or Explicit Everything) instead of Explicit method in Explicit + Implicit method?
>
> You are correct that this variation could be run, as there are many combination approaches that could be tried. However, it does not likely to change the outcome (since each of Explicit w/Decomp and Implicit do not perform better than No Decomp) and we already tried many novel approaches in our paper.  **Due to the combinatorial explosion of potential combinations of methods, the fact that no one uses those combinations, and the difficulty of displaying that many results, we did not include more than one combination approach.**
>
> ---
>
> **Presentation**: thank you for the feedback. We will update the colors and remove the word concurrent.
>
> **References**: We will add the reference, thank you.

---

### Official Review · Reviewer_mgx8 · 2023-08-04

**Soundness:** 4

**Excitement:**

3: Ambivalent: It has merits (e.g., it reports state-of-the-art results, the idea is nice), but there are key weaknesses (e.g., it describes incremental work), and it can significantly benefit from another round of revision. However, I won't object to accepting it if my co-reviewers champion it.

**Paper Topic And Main Contributions:**

The goal of this work is to understand if and when decompositions are useful in machine learning, as most research thus far has investigated open-domain.

**Questions For The Authors:**

See second point above.

**Reasons To Accept:**

This paper addresses an under-explored area in the QA/decompositions field.

**Reasons To Reject:**

* Motivation - what is the inherent difference of machine learning vs open-domain questions? What is the added value of studying specifically machine reading questions? For example, the question presented in Figure ` from HotpotQA can be both an open-domain question and a machine reading question (i.e. depending on whether context is included) -- I think an additional discussion explaining the different characteristic of the required reasoning would be helpful.
* Experiment details - I am not sure I understand the zero-shot setup in the size experiment as it is not explicitly stated, where do the decompositions come from at test time? Do the unified-QA/t5 models perform this decomposition in a zero-shot setting without seeing any such example? As this is a main result for the paper (full-data results with decompositions did not improve accuracy) this seems like a crucial point.
* Error analysis - since biggest category has invalid annotations, errors which are probably exhibited for both models, it would be more informative to examine only the set of errors where the fine-tuned succeeded and the zero-shot did not. At the moment it is hard for me to tell why does the zero-shot method perform worse.

**Reproducibility:**

4: Could mostly reproduce the results, but there may be some variation because of sample variance or minor variations in their interpretation of the protocol or method.

**Reviewer Confidence:**

4: Quite sure. I tried to check the important points carefully. It's unlikely, though conceivable, that I missed something that should affect my ratings.

---

> ### Author Rebuttal · Authors · 2023-08-28
>
> Thank you for your time and for recognizing that our work “addresses an under-explored area in the QA/decomposition field”!
>
> ---
> > What is the added value of studying specifically machine reading questions?
>
> The added value of studying machine reading (as opposed to ODQA) is that previous work has shown that decomposition helps in ODQA for improving retrieval. However, in MRC there is no retrieval so it is unclear if decomposition would help. We show that decompositions only helps when there’s not much data.
>
> Thus, we specifically do not study ODQA because we do not want to confound our experimental results with retrieval as that has been well-studied. We instead focus on studying the LMs and QA. **We agree this is important for the paper and do already include this discussion (of this motivation and definitions) in our paper on lines 33-45**.
>
> ---
>
> > where do the decompositions come from at test time? Do the unified-QA/t5 models perform this decomposition in a zero-shot setting without seeing any such example?
>
> **All decompositions, train and test, are provided by BREAK and are written by humans**. Thus, no model ever has to perform the decomposition as the decomposition is provided. We note that this means it is the oracle decomposition setting (e.g. gold decomposition at test time) and provides the strongest baseline for us to compare against. **Since No Decomp is still better or similar to them, this strengthens our claim that decompositions are not needed when you have enough training data**.
>
> Since the models just need to do regular QA (and not provide the decomposition themselves) it is fine for them to do standard QA and is a fair comparison. We note that the BREAK paper showed that model generated decompositions are worse than these human decompositions so we are using the best decompositions available.
>
> We will update lines 65-58 to make this more clear.
>
> ---
>
> > it would be more informative to examine only the set of errors where the fine-tuned succeeded and the zero-shot did not.
>
> We believe that our current error analysis shows the point we were hoping to make (that decompositions cause error propagations). **However, upon your suggestion we also ran your suggested error analysis and will add it to the appendix**.
>
> The new error analysis (where the fine-tuned No Decomp succeeded and the zero-shot Explicit Decomp failed, with 20 sampled errors) using UnifiedQAv2:
>
> | Reason               | Percent |
> |----------------------|---------|
> | Error Propagation    | 45%    |
> | Wrong Prediction at Last Step | 35%    |
> | Invalid or Missing Annotation   | 20%    |
>
> **We see that the results are similar to the previous error analysis in the paper**, as the No Decomp model was better than the Explicit case because of the lack of error propagation roughly 45% of the time (the current error analysis in the paper says 40% so they are similar).
>
> We hope this helps assuage your concern and will add this to the appendix to help future readers.

---

### Official Review · Reviewer_SCmL · 2023-08-11

**Soundness:** 4

**Excitement:**

3: Ambivalent: It has merits (e.g., it reports state-of-the-art results, the idea is nice), but there are key weaknesses (e.g., it describes incremental work), and it can significantly benefit from another round of revision. However, I won't object to accepting it if my co-reviewers champion it.

**Missing References:**

- For zero-shot decomposition, works like https://arxiv.org/abs/2205.10625 and https://arxiv.org/abs/2210.03350 have proposed prompting strategies that encourages the model to automatically decompose the question and answer itself, which are good to include in the related works.

**Paper Topic And Main Contributions:**

The paper aims to study whether or not question decomposition is helpful for machine reading comprehension tasks.

The paper conducted experiments on a few scenarios: explicit decomposition fine-tuning, implicit decomposition fine-tuning, and zero-shot explicit decomposition, compared against fine-tuning with varying amount of training data with no decomposition. The result from three models (bart, t5, unifiedqa-v2) on two datasets (DROP, HotpotQA) are collected.

The paper shows that incorporating decomposition in the fine-tuning process is either not helpful or detrimental. The zero-shot decomposition is outperformed by no decomposition fine-tuning with a few hundreds examples.

**Questions For The Authors:**

A. For fine-tuning with decomposition, are the test example also converted into corresponding decomposition format used in the training?

B. During training/evaluation, are context/supporting facts also passed to the model or only the questions are used?

**Reasons To Accept:**

- The paper is concise, the main claims are clear and can be routed to the supporting evidences easily.
- A good number of decomposition strategies are attempted, which further reinforces the claim of the paper.

**Reasons To Reject:**

- While I’m not expecting the author to conduct experiments on more powerful models, as the paper is making claim on the universal usefulness of decomposition, it’s good noting that for larger model the decomposition can work better (see Missing References). Performing error analysis on fine-tuning with decomposition can be helpful to understand which model capacity is essential for decomposition to work.

**Reproducibility:**

3: Could reproduce the results with some difficulty. The settings of parameters are underspecified or subjectively determined; the training/evaluation data are not widely available.

**Reviewer Confidence:**

3: Pretty sure, but there's a chance I missed something. Although I have a good feel for this area in general, I did not carefully check the paper's details, e.g., the math, experimental design, or novelty.

**Typos Grammar Style And Presentation Improvements:**

- In table 1, I think the error is wrong prediction at the first step instead of the last step?
- Showing an input-output pair for the decomposition fine-tuning strategies will be helpful for readers to understand what learning signal each method is contributing to the model.

---

> ### Author Rebuttal · Authors · 2023-08-28
>
> Thank you for your time and for recognizing that our work has “main claims [that] are clear and can be routed to the supporting evidences easily” and “a good number of decomposition strategies are attempted”!
>
> ---
>
> > While I’m not expecting the author to conduct experiments on more powerful models
>
> Previous work (“Is a Question Decomposition Unit All We Need?” by Patel et. al 2022 and others cited in our paper) have claimed to show the positive effects of decomposition for even standard size T5 models. Thus our experiments help shed more light on when decompositions are useful, using the same models previously studied.
>
> However, we agree that newer and bigger models are always helpful, so we have added Llama and Alpaca-Llama experiments on HotpotQA, which look similar to the T5 results. We can’t include a figure nicely in OpenReview, but here are the results in Table form (again, it has the same conclusion as the T5 results, e.g. No Decomp is better in all cases or within 2 SD otherwise). **This shows that even for newer models our findings hold: decompositions are helpful only when there are not enough labeled examples.**
>
> Scores are the mean and standard deviation of EM, like in the paper. Note that they are also better absolute scores than UnifiedQA-v2, despite only seeing the training set of HotpotQA (as opposed to the many QA datasets UnifiedQA-v2 trained on before further fine-tuning on HotpotQA train).
>
> Alpaca:
> |Decomposition Method|Scores            |
> |--------------------|------------------|
> |Explicit            |54.8 +- 0.4       |
> |Explicit w/Decomp   |53.8 +- 2.0         |
> |Explicit Everything |58.7 +- 1.3       |
> |Implicit            |57.0 +- 1.6         |
> |Implicit Unordered  |58.3 +- 0.7       |
> |Implicit w/Dupl.    |58.5 +- 2.1       |
> |Explicit & Implicit |55.0 +- 0.7         |
> |No Decomp           |59.9 +- 0.6       |
>
> Llama:
> |Decomposition Method|Scores            |
> |--------------------|------------------|
> |Explicit            |53.4 +- 3.5       |
> |Explicit w/Decomp   |57.7 +- 0.3       |
> |Explicit Everything |55.9 +- 0.5       |
> |Implicit            |60.9 +- 0.9       |
> |Implicit Unordered  |50.9 +- 0.6       |
> |Implicit w/Dupl.    |53.7 +- 0.6       |
> |Explicit & Implicit |59.3 +- 0.7       |
> |No Decomp           |59.8 +- 1.2       |
>
> We agree with unlimited time and compute, larger models (like 65B) would be better, but due to computational resources we could only do 7B models. We will add these tables to our appendix to further strengthen our claims.
>
> ---
> > A. For fine-tuning with decomposition, are the test example also converted into corresponding decomposition format used in the training?
>
> Yes, during test time, questions are also asked in the decomposition format (e.g. gold decompositions at test time). This provides the strongest baseline to compare to the No Decomp model, and despite it they are still similar or worse than No Decomp.
>
> > B. During training/evaluation, are context/supporting facts also passed to the model or only the questions are used?
>
> During training/evaluation, both contexts and questions are used, following previous work in MRC.
>
> ---
>
> **References**: we will make sure to add the missing reference. Thank you for pointing that out!
>
> **Presentation**: thank you for the suggestions, we will update the paper and add additional example input-output pairs (beyond Figure 1) to the appendix

---

### Meta-Review · Senior_Area_Chairs · 2023-10-01

**Recommendation:** 5

**Metareview:**

Thanks for your analysis. To me what you showed is kind of trivially true but there is a significant part of the research community who will find your results surprising. Therefore, I believe your paper can help to advance the state of the art in Question Answering.

---

### Decision · Program_Chairs · 2023-10-07

**Decision:**

Accept-Main

**Comment:**

Thanks for your analysis. To me what you showed is kind of trivially true but there is a significant part of the research community who will find your results surprising. Therefore, I believe your paper can help to advance the state of the art in Question Answering.